# WHICH RESTRAINS FEW-SHOT CLASS-INCREMENTAL LEARNING, FORGETTING OR FEW-SHOT LEARNING?

## ABSTRACT

Few-shot class-incremental learning (FSCIL) is one common yet difficult task in machine learning. There are mainly two challenges in FSCIL: catastrophic forgetting of old classes during incremental sessions and insufficient learning of new classes with only a few samples. Recent wisdom mainly focuses on how to avoid catastrophic forgetting by calibrating prototypes of each class while surprisingly overlooking the issue of limited samples of new classes. In this paper, we aim to improve the FSCIL by supplementing knowledge of new classes from old ones. To this end, we propose an old classes-guided FSCIL method with two stages of the base and incremental sessions. In the base session, we propose a prototype-centered loss trying to learn a compact distribution of old classes. During the incremental learning sessions, we first augment more samples for each new class by Gaussian sampling, where the mean and covariance are calibrated by old classes; we then propose to update the model based on both prototype-based and replay-based learning methods on those augmented samples. In addition, based on a series of analyses on examining the performance in both old and new classes during each session, we find out that most works contain a deceptive accuracy bias to old classes, where test data usually consists of a large part of samples in old classes. Extensive experiments on three popular FSCIL datasets including mini-ImageNet, CIFAR100, and CUB200 demonstrate the superiority of our model to the other state-of-the-art methods on both old and new classes.

## 1 INTRODUCTION

Trained on large datasets with a fixed number of classes, traditional deep neural networks are granted with high performance in recognizing all seen classes (Krizhevsky et al., 2012; He et al., 2016). However, the number of classes is usually non-stationary in real-world scenarios: new classes may appear after the deployment of the model. One straightforward way is to retrain the model using both old and new data, but old data is not always accessible due to safety or privacy regulations (Lesort et al., 2020) and fine-tuning the model only on the new data brings the well-known catastrophic forgetting problem (McCloskey & Cohen, 1989; Goodfellow et al., 2013). To tackle this issue, Class Incremental Learning (CIL) has attracted much attention, which aims to learn new concepts without forgetting old knowledge by simulating disjoint new classes emerging with session by session (van de Ven & Tolias, 2018; Zhu et al., 2021a;b; Masana et al., 2022). Furthermore, it is even more challenging when there are only a few samples for each new class during incremental sessions, which is usually called as Few-Shot Class Incremental Learning (FSCIL) (Tao et al., 2020; Tian et al., 2023).

**Motivation.** In general, FSCIL contains two stages where the model is firstly trained in a base session where all classes (i.e., old classes) contain enough samples, then continually learns few-shot new classes in each incremental session without access to the old data. After the training in each session, the model is evaluated on all seen classes so far. Previous studies (Tao et al., 2020; Cheraghian et al., 2021a; Akyürek et al., 2021; Song et al., 2023) find that the model usually suffers catastrophic forgetting of old classes during incremental learning sessions and is also vulnerable against overfitting on limited new data. Under such dilemma of the FSCIL, recent wisdom (Lesort et al., 2020; Zhu et al., 2021b; Hersche et al., 2022; Zhou et al., 2022a; Yang et al., 2023) proposes to design incremental learning strategies to suppress forgetting old classes and adapt smoothly to

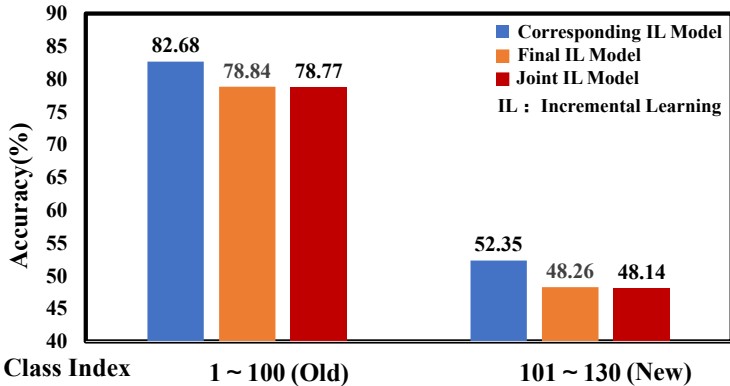

Figure 1: One FSCIL example to learn 30 new classes based on 100 old ones in the dataset CUB200. Blue bars show the average performance of the model learned in the corresponding session (each with 10 new classes), and orange bars show the performance of the final model after the last session, while red bars show the performance of the model learned in one joint session for all 30 new classes. We first can see that the accuracy of new classes is much lower than that of old classes. Furthermore, by comparing the performance of different models on new classes, we can conclude with two points. On the one hand, the performance of the final model (orange) is similar to the joint model (red), indicating that forgetting during incremental sessions is not a big issue. On the other hand, the corresponding IL model (blue) performs better than the joint IL model (red), demonstrating that more classes with few-shot samples bring difficulties.

new classes. However, few works shed light on the following question: *Which has a larger impact on FSCIL performance: the forgetting during incremental sessions or the few-shot data?*

To answer this question, we conduct an illustrative experiment under a prevailing FSCIL framework (Lesort et al., 2020; Hersche et al., 2022; Yang et al., 2023) in the dataset CUB200, where we learn 30 new classes based on 100 old ones by three incremental sessions (i.e., 10 new classes per session), which are shown in fig. 1. We can see that the performance of new classes is much lower than that of old classes, indicating the insufficient learning of new classes. Furthermore, we compare the performance of different models on the new classes, including models learned in the corresponding session (blue), the model learned in the last session (orange), and the model learned in one joint session for all 30 new classes (red). The difference between the final and joint model lies in learning incrementally or one-stage for those 30 new classes. On the one hand, we can see the performance of the final model is similar to the joint model, indicating that forgetting during incremental sessions is not a big issue. On the other hand, the corresponding IL model performs better than the joint IL model, demonstrating that more classes with few-shot samples bring difficulties (i.e., there are 130 classes in the joint IL model while only 110 and 120 classes in the model of the first and second session respectively.) In summary, these results suggest: *Under the simple incremental learning paradigm, FSCIL performance is more easily affected by the few-shot data.* Based on these observations, it is evident that if we aim to achieve better FSCIL performance, we should emphasize learning the few-shot data.

**Contribution.** In this paper, in order to enhance the model's ability to learn few-shot new classes, we propose to learn the model via two steps: firstly, we learn a base model with good generalization ability using a sufficient amount of old class data, then we augment data for new classes by sampling from a Gaussian distribution guided by the knowledge of old classes during incremental sessions. Concretely, for the base session where a sufficient amount of data is available, we propose a prototype-centered loss to facilitate better class separation, trying to push the features of each class to its corresponding prototypes, which learns more compact representations and boosts performance. During incremental learning sessions, we augment data for few-shot new classes by a Gaussian sampling, where the distribution of each new class is learned by a weighted combination of old classes and the weights are based on the similarity between the new class and old ones learned by an op-

timal transport framework. This data augmentation method can be flexibly adapted to most of the current FSCIL incremental learning strategies, such as prototype-based learning (Lesort et al., 2020; Zhu et al., 2021c; Hersche et al., 2022; Yang et al., 2023) or replay-based learning (Liu et al., 2022; Agarwal et al., 2022; Liu et al., 2023). For the prototype-learning, we calibrate the new prototype by weighting the contribution of each feature sampled from the new class distribution to mitigate the limited data issue, then update the projection layer to perform the alignment between the feature extractor and classifier (Hersche et al., 2022; Yang et al., 2023). For the replay-based method, we simply retrain the classifier on those augmented samples at each incremental session.

Our main contributions can be summarized as follows: (1) We empirically reveal the fact the current FSCIL task is more easily affected by the limited data issue instead of suffering catastrophic forgetting. (2) We propose a prototype-centered loss to enhance the model's generalization ability on the old classes, facilitating better class separation. (3) We propose an old class-guided method to learn sustainable new prototypes within an optimal transport framework, which enhances the model's ability to learn few-shot new classes and can be flexibly integrated into current FSCIL frameworks. (4) We conduct a comprehensive experimental analysis on FSCIL benchmark datasets, and the results show that our model substantially outperforms state-of-the-art methods on both old and new classes.

## 2 RELATED WORK

**Class Incremental Learning (CIL).** The fundamental goal of CIL is to adapt a pre-trained model of old classes to new classes without significantly deteriorating their performance in old ones. Recent research can be broadly divided into three categories to mitigate this issue. One of the most straightforward ways is to retain old data or knowledge. To maintain the old decision boundaries, previous retrained information is rehearsed during episodes of learning new classes (Rebuffi et al., 2017; Kang et al., 2022; Rolnick et al., 2019; Zhu et al., 2021a; Zhou et al., 2022b). Another common approach is to identify and freeze important model parameters dynamically and only update less important ones during incremental training (Kim et al., 2022a; Yan et al., 2021; Yoon et al., 2023; Li & Hoiem, 2017). The third category aims to correct the bias issue that the CIL methods usually are biased towards to those classes in the most recently learned sessions (Wu et al., 2019; Hou et al., 2019; Castro et al., 2018).

**Few-Shot Learning (FSL).** FSL aims to train models from a very limited number of examples. To generalize on the few-shot classes, metric-based approaches focus on learning a similarity metric that can distinguish between classes with few examples (Vinyals et al., 2016; Tian et al., 2020; Yu et al., 2022). Hallucination-based approaches (Wang et al., 2019; Yang et al., 2021; Chen et al., 2022; Guo et al., 2022) use data augmentation techniques, including geometric transformations and style transfer, have been explored to artificially increase the amount of training data. Furthermore, the weight generation-based methods (Gidaris & Komodakis, 2018; Qi et al., 2018) directly produce classification weights for new classes to mitigate the issue of overfitting.

**Few-Shot Class Incremental Learning (FSCIL).** FSCIL aims to learn new classes with a constraint of incoming data (Tao et al., 2020; Zhang et al., 2021; Zhou et al., 2022c;a; Yang et al., 2023). TOPIC (Tao et al., 2020) first defines this setting and utilizes a neural gas for topology preservation in the embedding space. To tackle the limited data issue in incremental sessions, prevailing studies adopted meta-learning approaches (Finn et al., 2017; Zhou et al., 2022c; Chi et al., 2022; Hersche et al., 2022) to simulate fake few-shot incremental episodes during the base training session. Nevertheless, meta-training approaches boost the model's generalization on old classes but hinder the model's ability to learn new classes. To overcome the catastrophic forgetting of old classes, recent studies (Zhang et al., 2021; Hersche et al., 2022; Yang et al., 2023) propose to freeze the feature extraction backbone after base training, fine-tuning a small number of extra parameters during incremental learning sessions. Zhu et al. (2021b) proposed a self-promoted prototype refinement mechanism to learn extensible feature representation in the base session. Unlike those methods, our approach directly calibrates the novel class prototypes using old class statics, allowing the prototypes to be separable but still within the close range of old prototypes. LDC (Liu et al., 2023) proposes to learn the new class distribution using an auto-regressive model and fine-tune the classifier using the sampled data from the learned distribution. Different from previous approaches, we integrate the new class distribution information into both prevailing prototype-based learning and replay-based learning methods.

## 3 METHODOLOGY

In this section, we first give a problem formulation of FSCIL, then describe our proposed FSCIL method via two stages of base and incremental session learning, respectively.

### 3.1 PROBLEM FORMULATION

FSCIL aims to train a classification model with $T$ sequential sessions $\left\{\mathcal{D}^{(0)}, \mathcal{D}^{(1)}, \ldots, \mathcal{D}^{(T)}\right\}$, where $\mathcal{D}^{(t)} = \{(\boldsymbol{x}_i^t, y_i^t)\}_{i=1}^{\left|\mathcal{D}^{(t)}\right|}$ is the training dataset at $t$ session. $\boldsymbol{x}_i^t$ is the input data and its label $y_i^t \in \mathcal{C}^t$. The label space $\mathcal{C}^t$ of dataset $\mathcal{D}^{(t)}$ is disjoint between different sessions, i.e., $\forall i \neq j, \mathcal{C}^{(i)} \cap \mathcal{C}^{(j)} = \emptyset$. The first session $\mathcal{D}^{(0)}$ is called the base session, which usually contains a sufficient amount of training data for each class $c \in \mathcal{C}^0$. In the following incremental session $\mathcal{D}^{(t)}$, there are $N$ new classes with $K$ training samples (usually 1 or 5 samples) in each class, formulating a $N$ way $K$ shot problem, i.e., $\left|\mathcal{D}^{(t)}\right| = N \cdot K$. At season $t$, previous datasets $\left\{\mathcal{D}^{(0)}, \mathcal{D}^{(1)}, \ldots, \mathcal{D}^{(t-1)}\right\}$ are not available, the model can only access to the data in $\mathcal{D}^{(t)}$. After training in season $t$, the model is evaluated on all seen classes $\tilde{\mathcal{C}}^t = \mathcal{C}^0 \cup \mathcal{C}^1 \ldots \cup \mathcal{C}^t$.

### 3.2 BASE SESSION LEARNING

Previous studies (Zou et al., 2022; Chi et al., 2022; Song et al., 2023) have revealed that pretraining in the base session profoundly affects the incremental learning of few-shot new classes. One good pre-trained base model should be able to separate each class while learning generalized class features for the incoming few-shot tasks (Kim et al., 2022b; Zou et al., 2022; Zhou et al., 2022a). Normally, the cross-entropy loss is adopted to train the base model by:

$$\mathcal{L}_{ce} = \mathbb{E}_{\boldsymbol{x}_i \sim \mathcal{D}^0} \left\{ \sum_{c=1}^{\mathcal{C}^0} -\log \frac{\mathbf{1}_{y_i=c}\left[\exp\left(\boldsymbol{w}_c^\top \phi\left(\boldsymbol{x}_i\right) + \boldsymbol{b}\right)\right]}{\sum_{c=1}^{\mathcal{C}^0} \exp\left(\boldsymbol{w}_c^\top \phi\left(\boldsymbol{x}_i\right) + \boldsymbol{b}\right)} \right\}, \tag{1}$$

where $\boldsymbol{w}_c$ represents the neuron weights in the liner layer for the $c$-th class, $\boldsymbol{b}$ is the bias, and $\phi(\cdot)$ is the backbone network. $\mathbf{1}_{(\cdot)}$ is the indicator function to determine whether the subscript condition is True, i.e., $\mathbf{1}_{True} = 1$, $\mathbf{1}_{False} = 0$. However, recent studies (Zou et al., 2022; Song et al., 2023) have demonstrated that such widely adopted cross-entropy training loss cannot efficiently separate class features.

To remedy this issue, we propose to learn more discriminative class features based on the learnable prototypes. Specifically, we set the classifier bias as zero and view neuron weights as prototypes, which are directly learned during the model training. Inspired by the supervised contrastive loss (Khosla et al., 2020), we propose a **Prototype-Centered Loss (PCL)** to push the extracted features of each class to each corresponding prototype:

$$\mathcal{L}_{pcl} = \mathbb{E}_{\boldsymbol{x}_i \sim \mathcal{D}^0} \left\{ \sum_{c=1}^{\mathcal{C}^0} -\log \frac{\mathbf{1}_{y_i=c}\left[\exp\left(cos\left(\boldsymbol{w}_c, \phi\left(\boldsymbol{x}_i\right)\right)\right)\right]}{\eta} \right\},$$

$$\eta = \sum_{c=1}^{\mathcal{C}^0} \exp\left(cos\left(\boldsymbol{w}_c, \phi\left(\boldsymbol{x}_i\right)\right)\right) + \sum_{\boldsymbol{x}_j \in \mathcal{D}^0, j \neq i} \mathbf{1}_{y_i \neq y_j} \exp\left(cos\left(\phi\left(\boldsymbol{x}_i\right), \phi\left(\boldsymbol{x}_j\right)\right)\right), \tag{2}$$

where $\boldsymbol{w}_c$ represents the prototype for the $c$-th class, $cos(\mathbf{x}, \mathbf{y}) = \frac{\mathbf{x}^\top \mathbf{y}}{\|\mathbf{x}\|\|\mathbf{y}\|}$ calculates the cosine similarity. We consider a positive sample if its class label is same as the prototype, otherwise negative samples. By minimizing $\mathcal{L}_{pcl}$, we gradually reduce the distance between each sample and its prototype while pushing apart the sample features from other classes. Finally, we combine both widely used cross-entropy loss and the proposed prototype-centered loss to be our final loss function by

$$\mathcal{L} = \mathcal{L}_{ce} + \mathcal{L}_{pcl}. \tag{3}$$

With the help of the prototype-centered loss, the intra-class features become more compactly clustered, and the inter-class features become more widely separated. The extra room in the feature embedding space benefits the incoming few-shot learning of new classes.

## 3.3 INCREMENTAL LEARNING SESSIONS

During incremental learning sessions $\{t|1 \leq t \leq T\}$, we first estimate a probability distribution of each new class calibrated from old classes in the $t$-th session, then we can augment more samples by sampling from each estimated distribution, finally we update the model by either the replay-based or prototype-based method. For the simplicity, we ignore the session index $t$ in the following description.

### 3.3.1 DISTRIBUTION ESTIMATION OF NEW CLASSES

Based on the empirical results as shown in fig. 1, we find that the FSCIL performance is more easily affected by the limited data issue. To tackle such issue of few-shot data, we aim to augment sufficient samples for each new class based on its corresponding probability distribution. However, it is challenging to estimate a distribution from only a few samples. On the other hand, there are usually enough samples for old classes during base session learning. Therefore, it is straightforward to think about estimating distribution of new classes guided from old classes.

We simply assume that the samples of each new class follow a Gaussian distribution where its mean and covariance are calibrated from old classes. To this end, we store the mean and covariance of each base class[1] to be used during the coming incremental sessions. For example, the mean and covariance of the $b$-th base class is calculated by

$$\boldsymbol{\mu}_b = \frac{1}{n_b}\sum_i^{n_b} \phi(\boldsymbol{x_i}), \qquad \boldsymbol{\Sigma}_b = \frac{1}{n_b-1}\sum_i^{n_b}(\phi(\boldsymbol{x_i})-\boldsymbol{\mu}_b)(\phi(\boldsymbol{x_i})-\boldsymbol{\mu}_b)^\top, \qquad (4)$$

where $n_b$ is the number of samples in $b$-th old class. For each few-shot new class, we first roughly calculated its mean as $\boldsymbol{\mu}_n = \frac{1}{K}\sum_i^K \phi(\boldsymbol{x_i})$, then update it by a weighted linear combination of means of old classes. To find the similarity weight between each base class to the new class, one intuitive approach is to use the cosine metric. However, we find cosine similarity brings two issues: (1) The disparity of weight range is not prominently evident, i.e., high similarity base classes are given relatively high weights while the non-similar classes are also given certain weights; (2) The possible negative similarity weights are likely to make the calibrated covariance not semi-positive definite. Thus, we propose to learn the similarity by minimizing a class-level optimal transport (OT) distance between old classes and the new class distribution with an entropic constraint (Cuturi, 2013):

$$\text{OT} = \sum_{b,n}^{\mathcal{B},N} \boldsymbol{C}_{b,n} \cdot \boldsymbol{T}_{b,n} - \epsilon\,\text{Entropy}(\boldsymbol{T}_{b,n}), \qquad (5)$$

$$\text{Entropy}(\boldsymbol{T}_{b,n}) = \sum_{b,n}^{\mathcal{B},N} -\boldsymbol{T}_{b,n} \cdot \ln \boldsymbol{T}_{b,n}, \qquad (6)$$

where $\epsilon$ is a hyperparameter, $\mathcal{B}, N$ represents the number of old and new classes respectively. $\boldsymbol{C} \in \mathbb{R}^{\mathcal{B} \times N}$ is the transport cost that indicates the distance between the new class $n$ and the base class $b$, here we use cosine distance $\boldsymbol{C}_{b,n} = 1 - cos(\boldsymbol{\mu}_b, \boldsymbol{\mu}_n)$. $\boldsymbol{T} \in \mathbb{R}^{\mathcal{B} \times N}$ is the positive transport probability matrix, which denotes the transport probability between the the $b$-th base class and the $n$-th new class. $\boldsymbol{T}_{b,n}$ provides a natural way to weigh the importance of each old class to new class, and it should satisfy the equipartition constraint $\left\{\sum_n^N \boldsymbol{T}_{bn} = \frac{1}{\mathcal{B}}, \sum_b^{\mathcal{B}} \boldsymbol{T}_{bn} = \frac{1}{N}\right\}$. With all learned similarity scores $\boldsymbol{T}_{bn}$, the calibrated mean and covariance of the $n$-th new class can be calculated by

$$\tilde{\boldsymbol{\mu}}_n = \frac{N \cdot \sum_{b\in\mathcal{B}} \boldsymbol{T}_{b,n}\boldsymbol{\mu}_b + \boldsymbol{\mu}_n}{\mathcal{B}+1}, \quad \tilde{\boldsymbol{\Sigma}}_n = \frac{N \cdot \sum_{b\in\mathcal{B}} \boldsymbol{T}_{b,n}\boldsymbol{\Sigma}_b}{\mathcal{B}} + \alpha, \qquad (7)$$

where $\alpha$ is a hyper-parameter that determines the degree of dispersion of covariance. In this way, we obtain a calibrated probability distribution for each new class, and then we can augment more samples to enhance the model's ability on few-shot new classes by the Gaussian sampling:

$$\delta_n = \left\{\delta|\delta \sim \mathcal{N}(\tilde{\boldsymbol{\mu}}_n, \tilde{\boldsymbol{\Sigma}}_n)\right\}, \delta_n \in \mathbb{R}^{M \times d}, \qquad (8)$$

---

[1]We will use the terms base class and old class interchangeably because, in this study, all old classes are learned during the base session.

where $\delta_n$ represents $M$ samples for the $n$-th new class, and $d$ denotes the feature dimension for each sample. In the following, we will introduce two methods to update our FSCIL models based on such augmented data.

### 3.3.2 REPLAY-BASED LEARNING

In replay-based learning, we directly retrain the classifier to align the backbone based on the augmented samples of all classes:

$$\mathbb{R}^{(t)} = \left\{ \delta_n^{(t)} | n = 0, 1, ..., N \right\}. \tag{9}$$

To avoid forgetting old classes, we also sample old classes features $\delta_0$ by using the base mean and covariance following the same way as new class sampling in eq. (8). In addition to such augmented samples, we add real few-shot samples of each new class to retrain the classifier by the cross-entropy loss:

$$\mathcal{L}_{Retrain} = \sum_{(\boldsymbol{x}, y) \sim \mathcal{D}^{(t)} \cup \mathbb{R}^{(t)}} - \log \Pr(y | \boldsymbol{x}, W^{(t)}), \tag{10}$$

where $W^{(t)}$ represents classifier parameters during the $t$-th session.

### 3.3.3 PROTOTYPE-BASED LEARNING

Prototype-based incremental learning methods have achieved much progress in FSCIL tasks (Lesort et al., 2020; Zhu et al., 2021c). Previous efforts focus on calibrating new prototypes by minimizing the similarity (Hersche et al., 2022; Yang et al., 2023) among all prototypes. Compared to traditional supervised learning, prototype learning requires less labeled data and offers stronger generalization abilities. However, with the few-shot data constraint, the calibrated prototypes of new classes are usually not accurate, which may damage the model's generalization.

In this study, we adopt all augmented samples to calibrate new prototypes. Though it can achieve good performance by directly using the calibrated mean in eq. (7) as the prototype of new class, we propose to learn the new prototype via a weighting scheme guided by old classes:

$$\boldsymbol{w_n} = \sum_{b,m} \boldsymbol{T}_{b,n} \boldsymbol{T}_{b,n,m} \boldsymbol{\delta}_{n,m}, \quad \boldsymbol{\delta}_{n,m} \in \mathbb{R}^d. \tag{11}$$

In the above, $\boldsymbol{T}_{b,n}$ represents the similarity score between $b$-th base class and the $n$-th new class, which is calculated by the class-level OT in eq. (6). $\boldsymbol{T}_{b,n,m}$ represents the similarity score between $b$-th base class and the $m$-th sample of the $n$-th new class, which can be optimized by a sample-level OT. The sample-level OT follows the same optimization as the class-level OT except that each distance/similarity is calculated based on a sample $\boldsymbol{\delta}_{n,m}$ instead of a mean vector $\boldsymbol{\mu}_n$.

Following previous works (Hersche et al., 2022; Yang et al., 2023), we divide the backbone network into one projection layer $g(\cdot)$ and one feature extractor $f(\cdot)$, i.e., $\phi(x) = g(f(x))$, and store the feature extractor output $\boldsymbol{h}$ for each class $c$:

$$\boldsymbol{H}^{(t)} = (\boldsymbol{h}_1, \boldsymbol{h}_2, \ldots, \boldsymbol{h}_{\tilde{\mathcal{C}}^t}), \quad \boldsymbol{h}_c = \frac{1}{|\mathcal{D}^{(t)}|} \sum_{i=1}^{|\mathcal{D}^{(t)}|} f\left(\boldsymbol{x}_i^t\right), \text{ s.t. } y_i^t = c, \quad 1 \le t \le T. \tag{12}$$

After obtaining the new prototypes in eq. (11), we retrain the projection layer to perform the alignment between the backbone and the classifier. In detail, we adopt a linear layer as the projection layer, and its parameters are optimized by minimizing a cosine similarity loss:

$$\mathcal{L}_{Align} = -\sum_c^{\tilde{\mathcal{C}}^t} cos(\boldsymbol{w}_c, g(\boldsymbol{h}_c)). \tag{13}$$

By minimizing such loss, we can obtain an optimized projector layer to align the backbone network and the classifier. During the test, we predict the input data $\boldsymbol{x}$ by comparing the similarity between backbone output $\phi(\boldsymbol{x})$ and the prototypes by

$$\underset{c}{\arg\max} \quad cos(\phi(\boldsymbol{x}), \boldsymbol{w}_c), c \in \left[1, \mathcal{C}^t\right]. \tag{14}$$

In summary, our proposed prototype-based method is shown in the following algorithm.

---

**Algorithm 1:** Prototype-based learning for each incremental session $t$.

---

**Input:** Few-shot new data $(\boldsymbol{x}^t, y) \in \mathcal{D}^{(t)}$, calibrated new distribution $\mathcal{N}(\tilde{\boldsymbol{\mu}}_n, \tilde{\boldsymbol{\Sigma}}_n)$ , previous classifier $W^{(t-1)}$.

**Output:** Linear projection layer $g(\cdot)$, classifier $W^{(t)}$.

1: Augment new samples $\boldsymbol{\delta}_{n,m}$ using $\mathcal{N}(\tilde{\boldsymbol{\mu}}_n, \tilde{\boldsymbol{\Sigma}}_n)$ in eq. (8);
2: Learn the similarity weight $\boldsymbol{T}_{b,n,m}$ by sample-level OT (similar with $\boldsymbol{T}_{b,n}$ eq. (6));
3: Calculate the new prototype $\boldsymbol{w_n}$ by eq. (11);
4: Obtain all prototypes $W^{(t)} = concat(W^{(t-1)}, \boldsymbol{w_n})$;
5: Obtain the feature extractor output features $\boldsymbol{H}^{(t)}$ from the input data $\boldsymbol{x^t}$ in eq. (12);
6: **for** *iterations* **do**

  Minimize alignment loss $\mathcal{L}_{Align}$ between prototypes $W^{(t)}$ and $g(\boldsymbol{h})$ in eq. (13);
  Backward;

---

# 4 EXPERIMENTS

## 4.1 IMPLEMENTATION DETAILS

We conduct extensive experiments on three benchmark datasets, including MiniImageNet, CI-FAR100, and CUB200, following previous works (Agarwal et al., 2022; Zhou et al., 2022c;a; Yang et al., 2023). For MiniImageNet and CIFAR, we set the number of base classes as 60. There are 8 Incremental sessions, and each session is a 5-way 5-shot (5 classes and 5 images per class) problem. For CUB200, we set the number of base classes as 100, followed by 10 incremental sessions and each session formulates a 10-way 5-shot problem. See more implementation details in appendix B.

## 4.2 BENCHMARK PERFORMANCE

We first report the average performance on CIFAR-100, minImageNet, and CUB-200, which are shown in table 1, table 2, and table 6, respectively. We compare our method with the current prototype-based and replay-based FSCIL methods according to (Tian et al., 2023). For the prototype-based method, we achieved the best performance in all sessions on CIFAR100, with at least 1.75% improvement in the last session. Although our prototype-based method did not surpass NC-FCIL and SAVC in the last session on MiniImageNet and CUB200 by a small margin, we still have the best average accuracy among all methods. For our simple replay-based method, we obtained the best performance in most sessions on all datasets with a substantial improvement. We further test the performance on the base and new classes at each session, which is shown in fig. 2. Our model maintains a good performance in the old classes while stably learning few-shot new classes.

Table 1: Incremental learning performance on CIFAR100 under 5-way 5-shot setup. "Avg Acc." represents the average accuracy of all sessions. "Final Improv." calculates the improvement of our method after learning in the final session.

| Methods | Accuracy in each session (%) ↑ | | | | | | | | | Avg Acc. | Final Improv. |
| --- | --- | --- | --- | --- | --- | --- | --- | --- | --- | --- | --- |
| | 0 | 1 | 2 | 3 | 4 | 5 | 6 | 7 | 8 | | |
| Self-promoted (Zhu et al., 2021c) | 64.10 | 65.86 | 61.36 | 57.45 | 53.69 | 50.75 | 48.58 | 45.66 | 43.25 | 54.52 | +14.61 |
| CEC (Lesort et al., 2020) | 73.07 | 68.88 | 65.26 | 61.19 | 58.09 | 55.57 | 53.22 | 51.34 | 49.14 | 59.53 | +8.72 |
| DSN Yang et al. (2022) | 73.00 | 68.83 | 64.82 | 62.64 | 59.36 | 56.96 | 54.04 | 51.57 | 50.00 | 60.14 | +7.86 |
| MetaFSCIL (Chi et al., 2022) | 74.50 | 70.10 | 66.84 | 62.77 | 59.48 | 56.52 | 54.36 | 52.56 | 49.97 | 60.79 | +7.89 |
| C-FSCIL (Hersche et al., 2022) | 77.47 | 72.40 | 67.47 | 63.25 | 59.84 | 56.95 | 54.42 | 52.47 | 50.47 | 61.64 | +7.39 |
| LIMIT (Zhou et al., 2022c) | 73.81 | 72.09 | 67.87 | 63.89 | 60.70 | 57.77 | 55.67 | 53.52 | 51.23 | 61.84 | +6.63 |
| ALICE (Peng et al., 2022) | 79.00 | 70.50 | 67.10 | 63.40 | 61.20 | 59.20 | 58.10 | 56.30 | 54.10 | 63.21 | +3.76 |
| SAVC (Song et al., 2023) | 78.77 | 73.31 | 69.31 | 64.93 | 61.70 | 59.25 | 57.13 | 55.19 | 53.12 | 63.63 | +4.74 |
| NC-FSCIL (Yang et al., 2023) | 82.52 | 76.82 | 73.34 | 69.68 | 66.19 | 62.85 | 60.96 | 59.02 | 56.11 | 67.50 | +1.75 |
| **Ours (Prototype-based learning)** | **83.43** | **78.92** | **75.00** | **70.89** | **67.88** | **64.84** | **62.57** | **60.48** | **57.86** | **69.10** | |
| iCaRL Rebuffi et al. (2017) | 64.10 | 53.28 | 41.69 | 34.13 | 27.93 | 25.06 | 20.41 | 15.48 | 13.73 | 32.87 | +43.57 |
| NCM (Hou et al., 2019) | 64.10 | 53.05 | 43.96 | 36.97 | 31.61 | 26.73 | 21.23 | 16.78 | 13.54 | 34.22 | +43.76 |
| Synthesized Replay (Cheraghian et al., 2021b) | 62.00 | 57.00 | 56.7 | 52.00 | 50.60 | 48.8 | 45.00 | 44.00 | 41.64 | 50.86 | +15.66 |
| Semantic Replay (Agarwal et al., 2022) | 70.14 | 64.36 | 57.21 | 55.21 | 54.34 | 51.89 | 50.12 | 47.91 | 46.61 | 55.31 | +10.69 |
| Data-free Replay (Liu et al., 2022) | 74.40 | 70.20 | 66.54 | 62.51 | 59.71 | 56.58 | 54.52 | 52.39 | 50.14 | 60.78 | +7.16 |
| **Ours (Replay-based learning)** | **83.43** | **78.91** | **74.56** | **69.99** | **66.66** | **63.74** | **62.13** | **59.87** | **57.30** | **68.51** | |

Table 2: Incremental learning performance on MiniImageNet under 5-way 5-shot setup. "Avg Acc." represents the average accuracy of all sessions. "Final Improv." calculates the improvement of our method after learning in the final session.

| Methods | Accuracy in each session (%) ↑ | | | | | | | | | Avg Acc. | Final Improv. |
|---|---|---|---|---|---|---|---|---|---|---|---|
| | 0 | 1 | 2 | 3 | 4 | 5 | 6 | 7 | 8 | | |
| Self-promoted (Zhu et al., 2021c) | 61.45 | 63.80 | 59.53 | 55.53 | 52.50 | 49.60 | 46.69 | 43.79 | 41.92 | 52.76 | +16.14 |
| CEC (Lesort et al., 2020) | 72.00 | 66.83 | 62.97 | 59.43 | 56.70 | 53.73 | 51.19 | 49.24 | 47.63 | 57.75 | +10.43 |
| Regularizer (Akyürek et al., 2021) | 80.37 | 74.68 | 69.39 | 65.51 | 62.38 | 59.03 | 56.36 | 53.95 | 51.73 | 63.71 | +6.33 |
| LIMIT (Zhou et al., 2022c) | 72.32 | 68.47 | 64.30 | 60.78 | 57.95 | 55.07 | 52.70 | 50.72 | 49.19 | 59.06 | +8.87 |
| MetaFSCIL (Chi et al., 2022) | 72.04 | 67.94 | 63.77 | 60.29 | 57.58 | 55.16 | 52.90 | 50.79 | 49.19 | 58.85 | +8.87 |
| C-FSCIL (Hersche et al., 2022) | 76.40 | 71.14 | 66.46 | 63.29 | 60.42 | 57.46 | 54.78 | 53.11 | 51.41 | 61.61 | +6.65 |
| LIMIT (Zhou et al., 2022c) | 73.81 | 72.09 | 67.87 | 63.89 | 60.70 | 57.77 | 55.67 | 53.52 | 51.23 | 61.84 | +6.83 |
| ALICE (Peng et al., 2022) | 80.60 | 70.60 | 67.40 | 64.50 | 62.50 | 60.00 | 57.80 | 56.80 | 55.70 | 63.99 | +2.36 |
| SAVC (Song et al., 2023) | 81.12 | 76.14 | 72.43 | 68.92 | 66.48 | 62.95 | 59.92 | 58.39 | 57.11 | 67.05 | +0.95 |
| NC-FSCIL (Yang et al., 2023) | 84.02 | 76.80 | 72.00 | 67.83 | 66.35 | 64.04 | 61.46 | 59.54 | **58.31** | 67.82 | -0.25 |
| **Ours (Prototype-based learning)** | **84.98** | **79.11** | **74.90** | **71.23** | **68.03** | **64.91** | **61.90** | **59.85** | 58.06 | **69.22** | |
| iCaRL Rebuffi et al. (2017) | 61.31 | 46.32 | 42.94 | 37.63 | 30.49 | 24.00 | 20.89 | 18.80 | 17.21 | 33.29 | +38.05 |
| NCM (Hou et al., 2019) | 61.31 | 47.80 | 39.30 | 31.90 | 25.70 | 21.40 | 18.70 | 17.20 | 14.17 | 30.83 | +41.09 |
| Synthesized Replay (Cheraghian et al., 2021b) | 61.40 | 59.80 | 54.20 | 51.69 | 49.45 | 48.00 | 45.20 | 43.80 | 42.10 | 50.63 | +13.16 |
| Semantic Replay (Agarwal et al., 2022) | 69.87 | 62.91 | 59.81 | 58.86 | 57.12 | 54.07 | 50.64 | 48.14 | 46.14 | 56.40 | +9.12 |
| Data-free Replay Liu et al. (2022) | 71.84 | 67.12 | 63.21 | 59.77 | 57.01 | 53.95 | 51.55 | 49.52 | 48.21 | 58.02 | +7.05 |
| **Ours (Replay-based learning)** | **84.98** | **77.60** | **74.16** | **70.37** | **67.46** | **63.13** | **60.08** | **57.75** | **55.26** | **67.87** | |

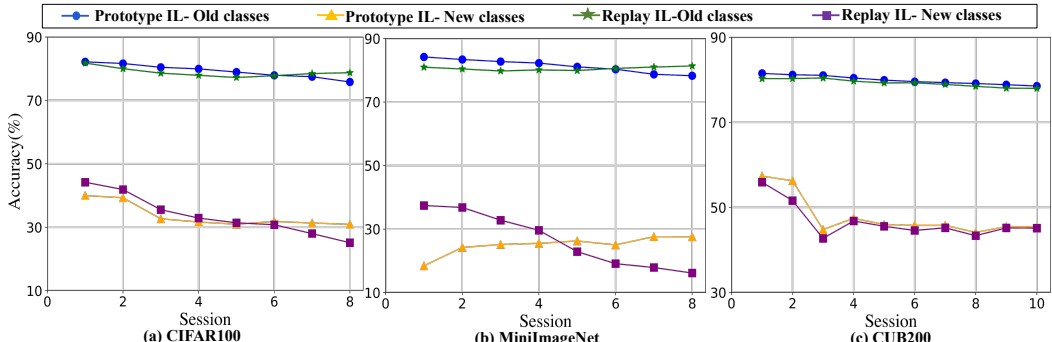

Figure 2: Performance of old classes and new classes on three benchmark datasets. We compare two different incremental learning methods, i.e., the blue circle and orange triangle denote the prototype incremental learning method, the green star and purple square represent the replay-based incremental learning method.

## 4.3 ABLATION STUDIES

**Base Session Learning.** We conduct the ablation experiments of prototype-centered loss. We compare the performance of the base model trained using cross-entropy loss (CE) and the model trained with prototype-centered loss. We also report the average incremental learning performance and the performance of the final session. As shown in table 3, prototype-centered loss boosts the performance of the base model, which allows the model to achieve better incremental learning and final performance. Especially for the fine-grained dataset CUB200, where different classes may share similar features, the improvement is more significant as prototype-center loss helps the model learn more discriminative features. We also visualize the features of both old and new classes in fig. 3 (see in appendix C). The CE-trained model cannot facilitate a good separation among classes, while the PCL-trained model learns more compact and separable representations.

**Incremental Learning Session.** We conduct the experiments using the mean feature of the few-shot classes as prototypes and calibrated new prototypes to validate the effectiveness of our proposed prototype-based learning method on CUB200. As shown in table 4, when only using the few-shot means (i.e., directly average the input few-shot data) as prototypes, the method obtains the lowest performance in all sessions. However, by using the statistically calibrated mean (i.e., $\tilde{\mu}_n$ in eq. (7)) as the prototype, the model achieved an average performance margin of 0.89%, which indicates the calibrated mean is more accurate, leading to better incremental performance. Our sampling method (i.e., $w_n$ in eq. (11)) demonstrates higher accuracy in most sessions, with the highest average

Table 3: Ablation results of the prototype-centered loss. We compare the performance of the model trained on base classes using only cross entropy (CE) and adding prototype alignment loss. "Base" is the performance after training the base session. "Average" indicates the average incremental learning performance. "Final" represents the final session performance.

| Method | CIFAR100 | | | MiniImageNet | | | CUB200 | | |
|---|---|---|---|---|---|---|---|---|---|
| | Base ↑ | Average↑ | Final↑ | Base↑ | Average↑ | Final↑ | Base↑ | Average ↑ | Final↑ |
| CE | 82.72 | 67.78 | 56.53 | 83.48 | 68.26 | 57.54 | 78.63 | 43.02 | 34.03 |
| CE+PAL | **83.43** | **69.10** | **57.86** | **84.98** | **69.17** | **57.86** | **82.68** | **70.05** | **61.81** |

accuracy of 70.05%. This result validates the effectiveness of our sampling and weighting strategy, which helps to learn better prototypes for the new classes. We further conduct the experiments in sampled numbers and add our method to the prevailing approaches to verify the effectiveness of our method (see in appendix D).

Table 4: Comparisons between using few-shot mean, calibrated mean and sampling-learned mean. "Few-shot prototype" denotes that we only use the few-shot data to calculate mean as prototypes. "Calibrated prototype" represents using the calibrated mean as prototypes (i.e., $\tilde{\mu}_n$ in eq. (7) ). "Sampling learned prototype " refers to the sampling and weighting learned prototypes (i.e., $w_n$ in eq. (11)).

| Methods | Accuracy In Each Session | | | | | | | | | | | Avg |
|---|---|---|---|---|---|---|---|---|---|---|---|---|
| | 0 | 1 | 2 | 3 | 4 | 5 | 6 | 7 | 8 | 9 | 10 | |
| Few-shot prototype | 82.68 | 78.23 | 75.42 | 71.75 | 69.61 | 67.19 | 65.56 | 64.22 | 62.29 | 61.74 | 60.73 | 69.04 |
| Calibrated prototype | 82.68 | 79.29 | 76.38 | 72.56 | 70.61 | 68.22 | 66.58 | 65.18 | 63.22 | 62.76 | **61.82** | 69.93 |
| Sampling learned prototype | **82.68** | **79.32** | **76.50** | **72.61** | **70.88** | **68.45** | **66.75** | **65.34** | **63.37** | **62.87** | 61.81 | **70.05** |

**Extending our method to prevailing FSCIL models.** We conduct additional experiments to integrate our approach with one popular incremental learning method C-FSCIL (Hersche et al., 2022) on CIFAR100. Specifically, during each incremental session, C-FSCIL maximizes the distances between old and new prototypes and fine-tunes the projection to perform the alignment between the feature extractor and the classifier. We utilize prototypes calibrated by distribution information and employing the same incremental learning methodology, and the results are shown in table 5. We observed a further improvement in the model's performance across the majority of sessions, ultimately achieving the highest average performance. These results demonstrate the capability of our method to complement existing approaches, thereby enabling the model to attain superior incremental learning performance.

Table 5: Experiments on integrating our method into another prototype-based incremental learning approach. We use the same base model while utilizing different incremental learning methods.

| Method | Accuracy in Each Session(%) | | | | | | | | | Avg |
|---|---|---|---|---|---|---|---|---|---|---|
| | 0 | 1 | 2 | 3 | 4 | 5 | 6 | 7 | 8 | |
| C-FSCIL (Hersche et al., 2022) | **83.43** | **79.02** | **75.11** | 69.88 | 66.91 | 63.60 | 61.82 | 59.52 | 56.50 | 68.42 |
| Ours+C-FSCIL | **83.43** | 78.92 | 74.99 | **70.95** | **67.81** | **64.81** | **62.70** | **60.54** | **57.94** | **69.12** |

## 5 CONCLUSION

In this paper, we find that FSCIL performance is more easily affected by the few-shot data instead of the forgetting. Motivated by this, we propose an old class-guided method to improve the FSCIL performance. First, we propose a prototype-centered loss to learn compact representations of the old classes during the base session learning. Next, we augment more samples for new classes by Gaussian sampling, where the probability distribution is calibrated from old classes by an optimal transport algorithm. Finally, we propose two methods to update the final model based on both replay-based and prototype-based learning on those augmented samples. Experimental results validate the superiority of our methods on three benchmark datasets on both old and new classes.

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

# A APPENDIX: SOTA COMPARISON ON CUB200

We reported the incremental learning performance on CUB200. For the prototype-based methods, although our method lagged behind by 0.69 in the final session, we achieved the highest performance in terms of average performance. For the replay-based methods, we obtain the same final performance as (Liu et al., 2023), achieving the best average performance.

Table 6: Performance of FSCIL in each session on CUB200 under 10-way 5-shot setup and comparison with other studies. "Average Acc." is the average accuracy of all sessions. "Final Improv." calculates the improvement of our method in the last session.

| Methods | Accuracy in each session (%) ↑ | | | | | | | | | | | Avg Acc. | Final Improv. |
|---|---|---|---|---|---|---|---|---|---|---|---|---|---|
| | 0 | 1 | 2 | 3 | 4 | 5 | 6 | 7 | 8 | 9 | 10 | | |
| CEC (Lesort et al., 2020) | 75.85 | 71.94 | 68.50 | 63.50 | 62.43 | 58.27 | 57.73 | 55.81 | 54.83 | 53.52 | 52.28 | 61.33 | +9.53 |
| LIMIT (Zhou et al., 2022c) | 76.32 | 74.18 | 72.68 | 69.19 | 68.79 | 65.64 | 63.57 | 62.69 | 61.47 | 60.44 | 58.45 | 66.67 | +3.36 |
| MetaFSCIL (Chi et al., 2022) | 75.9 | 72.41 | 68.78 | 64.78 | 62.96 | 59.99 | 58.3 | 56.85 | 54.78 | 53.82 | 52.64 | 61.93 | +9.17 |
| FACT (Zhou et al., 2022a) | 75.90 | 73.23 | 70.84 | 66.13 | 65.56 | 62.15 | 61.74 | 59.83 | 58.41 | 57.89 | 56.94 | 64.42 | +4.87 |
| ALICE (Peng et al., 2022) | 77.40 | 72.70 | 70.60 | 67.20 | 65.90 | 63.40 | 62.90 | 61.90 | 60.50 | 60.60 | 60.10 | 65.75 | +1.71 |
| NC-FSCIL (Yang et al., 2023) | 80.45 | 75.98 | 72.30 | 70.28 | 68.17 | 65.16 | 64.43 | 63.25 | 60.66 | 60.01 | 59.44 | 67.28 | +2.37 |
| SAVC (Song et al., 2023) | 81.85 | 77.92 | 74.95 | 70.21 | 69.96 | 67.02 | 66.16 | 65.30 | **63.84** | **63.15** | **62.50** | 69.35 | -0.69 |
| **Ours-Prototype-based learning** | **82.68** | **79.32** | **76.50** | **72.61** | **70.88** | **68.45** | **66.75** | **65.34** | 63.37 | 62.87 | 61.81 | **70.05** | |
| iCaRL (Rebuffi et al., 2017) | 68.68 | 52.65 | 48.61 | 44.16 | 36.62 | 29.52 | 27.83 | 26.26 | 24.01 | 23.89 | 21.16 | 36.67 | +40.42 |
| Synthesized Replay (Cheraghian et al., 2021b) | 68.78 | 59.37 | 59.32 | 54.96 | 52.58 | 49.81 | 48.09 | 46.32 | 44.33 | 43.43 | 43.23 | 51.84 | +18.35 |
| Data-free Replay (Liu et al., 2022) | 75.90 | 72.14 | 68.64 | 63.76 | 62.58 | 59.11 | 57.82 | 55.89 | 54.92 | 53.58 | 52.39 | 61.52 | +9.19 |
| LDC (Liu et al., 2023) | 77.89 | 76.93 | 74.64 | 70.06 | 68.88 | 67.15 | 64.83 | 64.16 | **63.03** | 62.39 | 61.58 | 68.32 | +0.00 |
| **Ours-Repaly-based learning** | **82.68** | **78.27** | **75.42** | **72.10** | **70.58** | **67.80** | **66.25** | **64.69** | 62.88 | 62.40 | 61.58 | **69.52** | |

# B APPENDIX: IMPLEMENTATION DETAILS

**Datasets.** We conduct experiments on three benchmark datasets, including CIFAR100 (Krizhevsky et al., 2009), MiniImageNet (Russakovsky et al., 2015) and CUB200 (Wah et al., 2011). MiniImageNet has 100 classes, each containing 500 images for training and 100 for testing with an image size of 84 × 84. CIFAR-100 has the same number of classes and images, and the image size is 32×32. CUB-200 is a dataset for fine-grained image classification containing 11,788 images of 200 classes in a resolution of 224× 224. There are 5,994 images for training and 5,794 images for testing. For MiniImageNet and CIFAR, we set the number of base classes as 60. Followed by 8 incremental sessions, each session is a 5-way 5-shot (5 classes and 5 images per class) problem. For CUB200, we set the number of base classes as 100, followed by 10 incremental sessions, each session formulates a 10-way 5-shot training data.

**Model Architecture.** Following previous works (Tao et al., 2020; Akyürek et al., 2021; Yoon et al., 2023), we use ResNet (He et al., 2016) as backbones. Following (Hersche et al., 2022; Yang et al., 2023), we use ResNet-12 in CIFAR100 and MiniImageNet experiments. Following (Yang et al., 2023; Song et al., 2023), we use ImageNet pre-trained ResNet-18 in CUB200 datasets. For the projection layer, we use a linear layer and set the dimension as 640 in implementation.

**Experimental Details.** For the base session, we train the model with 400 epochs on CIFAR100 and MiniImageNet with batch size 64; we train 350 epochs on CUB200 with batch size 256. During incremental sessions, for prototype-based learning, we train 50 iterations in the incremental sessions with batch size 64 on all datasets.For OT, the hyperparameter $\alpha$ and $\epsilon$ are set as 0.31 and 0.05 following (Yang et al., 2021) and (Wang et al., 2022), respectively. For replay-based learning, we train the model for 1000 iterations. The sample number $M$ in two incremental approaches is set as 400. We adopt a learning rate of 0.01 in all experiments and use SGD with momentum as an optimizer. We run each algorithm ten times for each dataset and report their mean accuracy. All experiments are conducted on one RTX 3090 graphics card. Our code will be publicly available in the final version.

# C APPENDIX: T-SNE VISUALIZATION OF PCL

We visualize the old class and new class data to demonstrate the effectiveness of PCL on CUB200. We visualize four old classes (class index 96 to 99) and two new classes (class index 100 and 101). It can be observed that the model trained using Cross-Entropy (CE) is unable to effectively distinguish

between the old and new categories, and there is an overlap of new class points and old classes in the high-dimensional space. On the other hand, the model trained using PCL separates each class widely and learns compact features, thus providing a solid foundation for subsequent few-shot incremental learning.

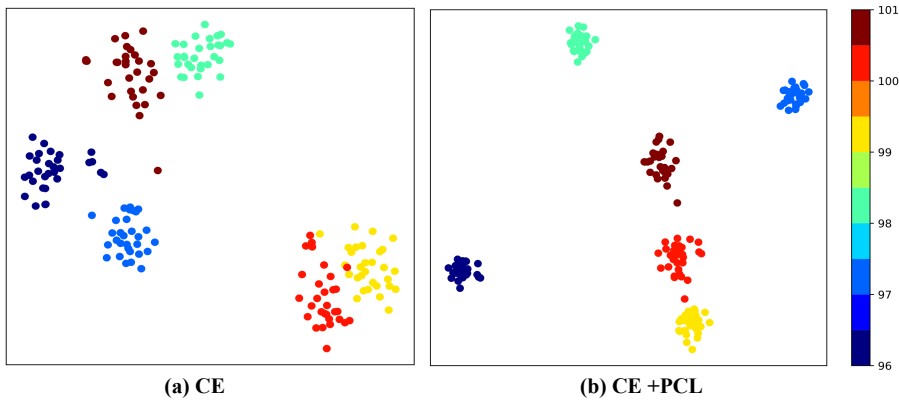

(a) CE          (b) CE +PCL

Figure 3: T-sne visualization of the old and new classes on CUB200. Red and brown points are new class data (class index 100 and 101), and the rest are old class data. The CE-trained model cannot facilitate a good separation of classes, as the new classes overlap with each other. The PCL-trained model facilitates a better separation.

# D    APPENDIX: ABLATION RESULTS IN INCREMENTAL SESSION LEARNING

**Effect of Sample Numbers.** We conducted sensitivity experiments varying the number of sampling points. We conduct experiments on the number of sampling features varying from 100 to 1000 using two incremental learning approaches. As shown in fig. 4, it is evident from the graph that the performance on the CIFAR100 dataset remains relatively stable and is minimally impacted by the number of sampling points. To balance the training speed and the performance, we chose 400 as the number of sampled features.

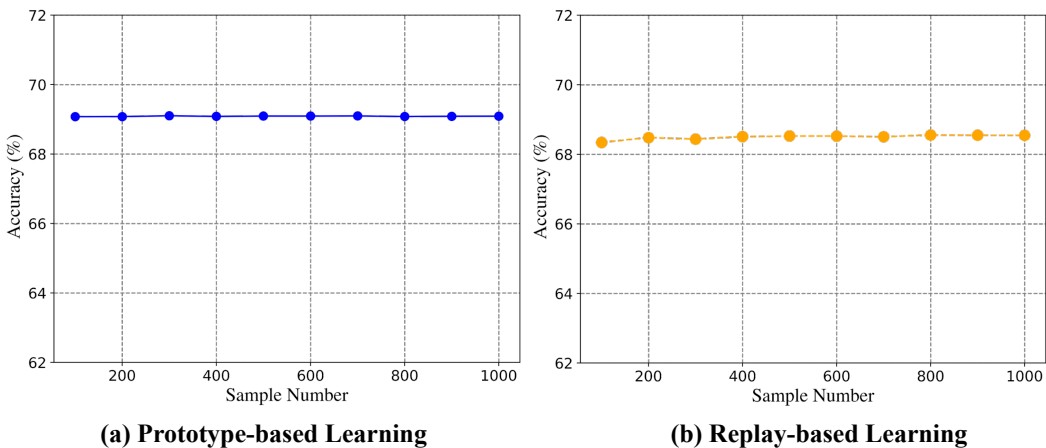

(a) Prototype-based Learning          (b) Replay-based Learning

Figure 4: The effect of different values of sampled numbers. We conduct the ablation experiments on CIFAR100. (a) The effect of sampling numbers in prototype-based learning during incremental sessions. (b) The effect of sampling numbers in replay-based learning during incremental sessions.

