# OpenReview forum: "WHICH RESTRAINS FEW-SHOT CLASS-INCREMENTAL LEARNING, FORGETTING OR FEW-SHOT LEARNING?"
_ICLR.cc/2024/Conference — ICLR 2024 Conference Withdrawn Submission_

### Official Review · Reviewer_WPjb · 2023-10-22

**Soundness:** 2 fair
**Presentation:** 2 fair
**Contribution:** 2 fair
**Rating:** 3
**Confidence:** 5

**Summary:**

This paper tackles the few-shot class-incremental learning problem, which is of great importance to the machine learning field. The authors propose to analyze the restraint for few-shot class-incremental learning and find the core problem is the performance of new classes. They propose a sampling process to sample new class features to help the model learn new classes better. The proposed method is evaluated on several benchmarks against other baselines.

**Strengths:**

1. This paper tackles the few-shot class-incremental learning problem, which is of great importance to the machine learning field.
2. The source code is attached, enabling reproducing the results (although I did not check the reproducibility)
3. The proposed method is evaluated on several benchmarks against other baselines.

**Weaknesses:**

1. First of all, I must highlight that the claim point in the introduction (Figure 1) may not be true for all cases. In the introduction, the authors evaluate the performance of old and new classes on several methods and find the accuracy of new classes is much lower than that of old classes. However, this claim point may be biased since (almost) all few-shot class-incremental learning methods tend to freeze the backbone and extract prototypes for new classes (now a common practice). Hence, the old classes’ performance will be maintained as the model is frozen, restricting its plasticity. We can assume a typical baseline to update the model without freezing any parameters with only few-shot instances, where we definitely will find the accuracy of new classes will be larger than old ones due to catastrophic forgetting. In other words, the analysis in Figure 1 may not be a sufficient enplanement for the conclusion that “few-shot instances restrain few-shot class-incremental learning”.
2. Secondly, the main idea of this paper is somehow incremental due to the overlapping of another typical work in the few-shot learning field [1]. In [1], the authors find similar classes have similar mean and variance. Hence, they utilize the similarity-based weight to calibrate the distribution of new classes with old classes. The core idea is almost the same as the current method (which only changes the similarity-based calibration into OT-based calibration). Hence, highlighting the difference to [1] requires careful discussion and ablation. I would also like to see the results when using [1] to sample new class features in the current method.
3. In my opinion, Eq. 7 calibrates the prototypes of new classes with old classes and helps the learning of new classes. However, it still has a possible negative impact on the old classes. Taking a 2-class classification problem for an example, if you are moving the prototype of class 2 nearer to class 1, the classification accuracy of class 1 will correspondingly decrease since the new prototype pushes the decision boundary. Discussing such negative impacts and why the final performance will improve would be better. Additionally, showing the accuracy of old and new classes will clarify this.
4. The performance comparison in Table 2 shows that the main improvement comes from the base task training. Specifically, compared to NC-FSCIL, “Ours (Prototype-based learning)” has better performance in the former stages while having lower performance in the latter stages. Hence, the main improvement in the “Avg Acc” column comes from the improvement of the base session instead of the capability to resist forgetting. Such a learning protocol seems to avoid the inherent goal of class-incremental learning, leading to biased works that work on pre-training the model with complex tricks instead of designing useful training techniques.

**Questions:**

Please also address these concerns in the rebuttal.
1. Ablations are needed for the class calibration. Instead of using OT, other similarity-based calibration can also be applied, e.g., [1].
2. In the experimental results, some compared methods are using resnet20 while others are using resnet18. Please specify these differences clearly in the tables.
3. Why does replay show inferior performance than Prototype-based learning, although it takes more memory? Discussions are needed.
4. There are two LIMIT in Table 2, with different performances. What is the reason for this?
5. Some clarifications are needed for the inference process. In Eq. 3, cross-entropy and prototype loss are added. However, the bias term is removed for the prototype loss. Does that mean the cross entropy is done without the bias term? Which is utilized for the final inference? Some clarifications are needed.
6. According to the fourth aforementioned weakness, it would be better to align the performance of different methods at the base session and compare the performance to resist forgetting in the incremental learning stage.

[1] Free Lunch for Few-shot Learning: Distribution Calibration. ICLR 2021

In summary, this paper tackles an interesting problem. My concerns are mainly based on motivation, incremental contribution, unfair comparison, and blurry negative impacts.

---

### Official Review · Reviewer_VZnS · 2023-11-01

**Soundness:** 2 fair
**Presentation:** 2 fair
**Contribution:** 2 fair
**Rating:** 3
**Confidence:** 5

**Summary:**

To improve Few-shot class-incremental learning (FSCIL), the paper proposed a prototype-centered loss trying to encourage a compact distribution of old classes during the base session, and for incremental learning sessions, it uses prototype-based and replay-based learning on the calibrated augmented samples based on Gaussian sampling. Empirical improvement is observed on mini-ImageNet, CIFAR100, and CUB200 datasets.

**Strengths:**

The paper provides thorough experiments on various datasets including mini-ImageNet, CIFAR100, and CUB200, along with ablation studies on the proposed components.

**Weaknesses:**

It is understandable that "...performance of new classes is much lower than that of old classes..." (page 2) because the images of the old classes significantly outnumber the images of the new classes. Hence, this observation is somewhat trivial.

The main contribution is exactly the same as "Distribution Calibration (DC)" which has already been widely adopted in few-shot learning. If this work simply borrows the same idea from few-shot learning to FSCIL, then the contribution is quite limited.

**Questions:**

There are two settings adopted in the paper, i.e., Prototype-based learning and Replay-based learning. However, the Replay-based CIL is a common setting for many-shot CIL. In FSCIL, how is replay performed, especially when there is only a single one-shot example available?

---

### Official Review · Reviewer_mFn2 · 2023-11-06

**Soundness:** 3 good
**Presentation:** 3 good
**Contribution:** 3 good
**Rating:** 6
**Confidence:** 3

**Summary:**

The paper addresses the challenges in few-shot class-incremental learning (FSCIL), which involves adding new classes to a model with limited examples without forgetting the previously learned classes. The authors argue that the main issues in FSCIL are not just catastrophic forgetting of old classes but also the insufficient learning of new classes due to the few-shot scenario. They propose a two-stage method to enhance FSCIL by first learning a compact distribution of old classes and then augmenting the new class data by Gaussian sampling informed by the old classes. The paper introduces a prototype-centered loss for the base session to learn compact class representations and a method to augment data for new classes during incremental learning sessions. They also highlight a common bias in FSCIL models towards old classes and demonstrate the effectiveness of their method through experiments on standard datasets.

**Strengths:**

1. The paper presents an innovative approach by focusing on the few-shot learning aspect of FSCIL, which is often overshadowed by the problem of catastrophic forgetting. The introduction of a prototype-centered loss and the use of old class data to inform the distribution of new class data are novel contributions to the field.
2. The proposed method is well-grounded in empirical analysis, and the authors provide a convincing argument supported by experiments. The quality of the research is evident from the methodological rigor and the thoroughness of the experimental setup.
3. The paper is clearly written, with a well-defined problem statement, a detailed explanation of the proposed method, and a logical flow of ideas. The use of figures to illustrate the problem and the results adds to the clarity of the presentation.

**Weaknesses:**

1. The paper could strengthen its claims by testing the method across a wider variety of datasets and incremental learning scenarios to ensure the generalizability of the results.
2. While the paper compares the proposed method with state-of-the-art approaches, a more detailed analysis of how the method performs relative to a broader range of existing methods could provide a clearer picture of its advantages and limitations.

**Questions:**

1. Could the authors elaborate on the choice of Gaussian sampling for data augmentation and whether other distributions were considered or would be applicable?
2. The paper mentions a bias towards old classes in FSCIL models. Could the authors discuss potential strategies to mitigate this bias further?
3. How does the proposed method scale with the number of classes and the size of the datasets? Are there computational constraints that need to be considered?
4. Is there potential for the proposed method to be integrated with other FSCIL strategies, and if so, what might be the expected benefits or trade-offs?